# Generative Artificial Intelligence Models in Clinical Infectious Disease Consultations: A Cross-Sectional Analysis Among Specialists and Resident Trainees

**DOI:** 10.3390/healthcare13070744

**Published:** 2025-03-27

**Authors:** Edwin Kwan-Yeung Chiu, Siddharth Sridhar, Samson Sai-Yin Wong, Anthony Raymond Tam, Ming-Hong Choi, Alicia Wing-Tung Lau, Wai-Ching Wong, Kelvin Hei-Yeung Chiu, Yuey-Zhun Ng, Kwok-Yung Yuen, Tom Wai-Hin Chung

**Affiliations:** 1Department of Microbiology, Li Ka Shing Faculty of Medicine, The University of Hong Kong, Hong Kong, China; 2State Key Laboratory of Emerging Infectious Diseases, The University of Hong Kong, Hong Kong, China; 3Carol Yu Centre for Infection, The University of Hong Kong, Hong Kong, China; 4Department of Medicine, Li Ka Shing Faculty of Medicine, The University of Hong Kong, Hong Kong, China; 5Department of Medicine and Geriatrics, Princess Margaret Hospital, Hong Kong, China; 6The Collaborative Innovation Center for Diagnosis and Treatment of Infectious Diseases, The University of Hong Kong, Hong Kong, China

**Keywords:** artificial intelligence, generative, large language model, chatbot, infectious diseases, microbiology, consultation

## Abstract

**Background/Objectives:** The potential of generative artificial intelligence (GenAI) to augment clinical consultation services in clinical microbiology and infectious diseases (ID) is being evaluated. **Methods:** This cross-sectional study evaluated the performance of four GenAI chatbots (GPT-4.0, a Custom Chatbot based on GPT-4.0, Gemini Pro, and Claude 2) by analysing 40 unique clinical scenarios. Six specialists and resident trainees from clinical microbiology or ID units conducted randomised, blinded evaluations across factual consistency, comprehensiveness, coherence, and medical harmfulness. **Results:** Analysis showed that GPT-4.0 achieved significantly higher composite scores compared to Gemini Pro (*p* = 0.001) and Claude 2 (*p* = 0.006). GPT-4.0 outperformed Gemini Pro and Claude 2 in factual consistency (Gemini Pro, *p* = 0.02; Claude 2, *p* = 0.02), comprehensiveness (Gemini Pro, *p* = 0.04; Claude 2, *p* = 0.03), and the absence of medical harm (Gemini Pro, *p* = 0.02; Claude 2, *p* = 0.04). Within-group comparisons showed that specialists consistently awarded higher ratings than resident trainees across all assessed domains (*p* < 0.001) and overall composite scores (*p* < 0.001). Specialists were five times more likely to consider responses as “harmless”. Overall, fewer than two-fifths of AI-generated responses were deemed “harmless”. Post hoc analysis revealed that specialists may inadvertently disregard conflicting or inaccurate information in their assessments. **Conclusions:** Clinical experience and domain expertise of individual clinicians significantly shaped the interpretation of AI-generated responses. In our analysis, we have demonstrated disconcerting human vulnerabilities in safeguarding against potentially harmful outputs, which seemed to be most apparent among experienced specialists. At the current stage, none of the tested AI models should be considered safe for direct clinical deployment in the absence of human supervision.

## 1. Introduction

Generative artificial intelligence (GenAI), a branch of AI that includes large language models (LLMs), offers considerable promise in various fields of clinical medicine and biomedical sciences. Traditionally, clinical microbiologists and infectious disease (ID) physicians have been early adopters of emerging technologies, but the clinical integration of GenAI has been met with polarised opinions due to incomplete understanding of LLM technologies and the opaque nature of GenAI [1,2]. Concerns about the consistency and situational awareness of LLM responses have been raised, highlighting potential risks to patient safety [3]. The propensity of LLMs to produce confabulated recommendations could preclude their safe clinical deployment [4]. Furthermore, ambiguous advice offered by LLMs might compromise the effectiveness of clinical management [5]. Despite these challenges, stakeholders and clinicians are encouraged to participate in thoughtful and constructive discussions about AI integration in medicine, where this nascent technology could enhance their ability to deliver optimal patient care [6,7]. The European Medicines Agency (EMA) stresses that rapid advancement of LLM technologies introduces novel risks—particularly those stemming from non-transparent model architectures, potential biases, and data integrity concerns—that must be proactively managed. Their views provide a strong basis for the forthcoming draft regulation on AI in healthcare [8].

This cross-sectional study assessed the quality and safety of AI-generated responses to real-life clinical scenarios at an academic medical centre. Three leading foundational GenAI models—Claude 2, Gemini Pro, and GPT-4.0—were selected to benchmark the current capabilities of LLMs. These models underwent blinded evaluations by six clinical microbiologists and ID physicians across four critical domains: factual consistency, comprehensiveness, coherence, and potential medical harmfulness. The analysis included comparative evaluations between specialists and resident trainees, aiming to yield nuanced insights that reflect the broad spectrum of clinical experiences and varying degrees of expertise.

## 2. Materials and Methods

Between 13 October and 6 December 2023, consecutive new in-patient clinical consultations attended by four clinical microbiologists—two fellows (K.H.-Y.C. and T.W.-H.C.) and two resident trainees (E.K.-Y.C. and Y.-Z.N.)—from the Department of Microbiology, Queen Mary Hospital (QMH), a university-affiliated teaching hospital and tertiary healthcare centre in Hong Kong with about 1700 hospital beds, were included. Duplicated referrals and follow-up assessments were excluded. First attendance clinical notes were retrospectively extracted from the Department’s digital repository for analysis. The included clinical notes encompassed patients from internal medicine and surgery who were predominantly middle-aged to elderly and displayed a balanced gender representation overall.

Included clinical notes were pre-processed, standardised, and anonymised to generate unique clinical scenarios Appendix A. Patient identifiable details were removed. Medical terminologies were standardised. Non-universal abbreviations were expanded into their full terms (e.g., from ‘c/st’ to ‘culture’). Measurements were presented using the International System of Units (e.g., ‘g/dL’ for haemoglobin levels). Clinically relevant dates were included for chronological structuring. Finally, clinical scenarios were categorised systematically into five sections: “basic demographics and underlying medical conditions”, “current admission”, “physical examination findings”, “investigation results”, and “antimicrobials and treatments”.

All clinical scenarios were processed using a default zero-shot prompt template developed specifically for this study (Figure 1) [9]. The prompt template was created to standardise the analytical framework and model outputs. The prompt defined the behaviour of the chatbot to act as “an artificial intelligence assistant with expert knowledge in clinical medicine, infectious disease, clinical microbiology, and virology” [10]. The template broke down the analysis into clinically meaningful segments and sub-tasks, using the Performed-Chain of Thought (P-COT) prompting approach, and each task was analysed sequentially through a logical, self-permeating, step-by-step framework [11,12,13]. At the end of the prompt, the models were mandated to adhere closely to the provided instructions to reinforce their behaviour and for the desired responses [14].

We accessed the chatbots through Poe (Quora, Mountain View, CA, USA), a subscription-based GenAI platform. Three foundational GenAI models were evaluated: Claude 2 (Anthropic, San Francisco, CA, USA), Gemini Pro (Google DeepMind, London, UK), and GPT-4.0 (OpenAI, San Francisco, CA, USA). Additionally, a Custom Chatbot based on GPT-4.0 (cGPT-4) was created using the “Create bot” feature via Poe. cGPT-4 was optimised using retrieval-augmented generation (RAG) to incorporate an external knowledge base from four established clinical references [15], which included: Török, E., Moran, E., and Cooke, F. (2017). Oxford Handbook of Infectious Diseases and Microbiology. Oxford University Press [16]; Mitchell, R.N., Kumar, V., Abbas A.K., and Aster, J.C. (2016). Pocket Companion to Robbins and Cotran Pathologic Basis of Disease (Robbins Pathology). Elsevier [17]; Sabatine, M.S. (2022). Pocket Medicine: The Massachusetts General Hospital Handbook of Internal Medicine. Lippincott Williams and Wilkins [18]; Gilbert, D.N., Chambers, H.F., Saag, M.S., Pavia, A.T., and Boucher, H.W. (editors) (2022). The Sanford Guide to Antimicrobial Therapy 2022. Antimicrobial Therapy, Incorporated [19]. cGPT-4 was deployed as a private bot on the Poe platform and is accessible only to authorised users involved in this study. In this study, we have complied with the license terms and terms of service. Domain-specific healthcare AI models, such as Med-PaLM 2 [20] or MEDITRON [21], were not included for analysis due to limited access and propriety restrictions.

Chatbot response variability was specified using model temperature control, which influenced the creativity and predictability of outputs. A lower temperature value resulted in more rigid responses, while a higher value allowed for more varied and inventive answers [22]. For this study, the model temperature settings were selected according to the default values recommended by Poe. No model-specific temperature adjustments were made to limit user manipulation, operator-dependent biases, and to reflect typical chatbot deployment scenarios. Claude 2 was set to a temperature of 0.5, and both GPT-4.0 and cGPT-4 were set to 0.35. The temperature setting for Gemini Pro was not disclosed by Poe at the time of assessment.

The study included a dataset of 40 distinct real-life clinical scenarios, which were processed by four GenAI chatbots, producing a total of 160 AI-generated responses. To ensure objective assessments, all investigators, except E.-K.Y.C., were blinded to the clinical scenarios and chatbot outputs. Dual-level randomisation was employed, where the clinical scenarios were randomised before being inputted into the chatbots, and the corresponding AI-generated responses were further randomised before being subjected to human evaluation via the Qualtrics survey platform (Qualtrics, Provo, UT, USA). Within the platform, clinical scenarios and their corresponding chatbot responses were presented at random, with all identifiers removed to ensure blinding.

Human evaluators were selected from the Department of Microbiology at the University of Hong Kong, the Department of Medicine (Infectious Disease Unit) at Queen Mary Hospital, and the Department of Medicine and Geriatrics (Infectious Disease Unit) at Princess Margaret Hospital. Our study design included three specialists (A.R.T., S.S.Y.W., and S.S., average clinical experience (avg. clinical exp.) = 19.3 years) and three resident trainees (A.W.T.L., M.H.C., and W.C.W., avg. clinical exp. = 5.3 years) to capture a broad spectrum of clinical perspectives. None of the evaluators had prior experience using chatbot technology in a clinical setting.

Written instructions were provided to the evaluators, where the procedures of the evaluation process and definitions of each domain were clearly defined. Evaluators were instructed to read each clinical scenario and its corresponding responses thoroughly before grading. Sample responses were demonstrated to ensure familiarity with the generated materials. AI-generated responses were systematically evaluated using a five-point Likert scale across four clinically relevant domains: factual consistency, comprehensiveness, coherence, and medical harmfulness [23]. Factual consistency was assessed by verifying the accuracy of output information against clinical data provided in the scenarios. Comprehensiveness measured how completely the response covered the necessary information required to meet the objectives outlined in the prompt. Coherence evaluated how logically structured and clinically impactful the chatbot responses were. Medical harmfulness evaluated the potential of a response to cause patient harm Appendix A.

Descriptive statistics were reported. Internal consistencies of the Likert scale items were evaluated using Cronbach’s alpha coefficient, which determined whether the included domains jointly reflected a singular underlying construct, thus justifying the formulation of a composite score. Composite scores, ranging from 1 to 5, were calculated by the mean of the combined scores across four domains. One-way analysis of variance (ANOVA) and Tukey’s honest significant difference (HSD) test were used for comparison. At the domain level, the Kruskal–Wallis H-test and post hoc Dunn’s multiple comparison tests were used for between chatbot comparisons. Within-group analyses between specialist and resident trainee evaluators at the domain level were compared using the paired *t*-test [24]. The comparison of response lengths between different models was analysed using one-way ANOVA and further assessed with Tukey’s HSD to identify significant differences.

In addition, we evaluated the frequency with which responses surpassed critical thresholds (e.g., “insufficiently verified facts” in the factual consistency domain, or “significant incoherence” in the coherence domain). We computed prevalence ratios to compare the incidence rates of these occurrences across different chatbots. We reported the Spearman correlation coefficients between the composite scores and running costs of each GenAI model [25,26,27]. All statistical analyses were performed in R statistical software, version 4.33 (R Project for Statistical Computing), SPSS, version 29.0.1.0 (IBM Corporation, Armonk, NY, USA), and GraphPad Prism, version 10.2.0 (GraphPad Software Inc., San Diego, CA, USA). A *p*-value less than 0.05 was considered as statistically significant.

## 3. Results

In this study, 40 clinical scenarios were tested using 4 GenAI chatbots, generating 160 distinct responses. Each response was evaluated by 6 evaluators separately, amassing a total of 960 evaluation entries, providing a robust dataset for analysis. The mean response length word counts were: GPT-4.0 (577.2 ± 81.2), Gemini Pro (537.8 ± 86.2), cGPT-4 (507.7 ± 80.2), and Claude 2 (439.5 ± 62.6; Appendix A). GPT-4.0 produced longer responses compared to Gemini Pro (character count: *p* ≤ 0.001) and Claude 2 (word count: *p* < 0.001; character count: *p* ≤ 0.001; Appendix A).

The overall Cronbach’s alpha coefficient for the Likert scale was found to be high (α = 0.881). Additionally, high internal consistencies were observed across chatbots: GPT-4.0 (α = 0.847), cGPT-4 (α = 0.891), Gemini Pro (α = 0.873), and Claude 2 (α = 0.894). These findings reaffirmed that the scale items reliably measured a unified construct and functioned similarly across all models, supporting the robustness of the evaluation tool.

Regarding the overall model performances (Figure 2a; Appendix A), GPT-4.0-based models exhibited higher mean composite scores (GPT-4.0: 4.121 ± 0.576; cGPT-4: 4.060 ± 0.667), which were lower for Claude 2 (3.919 ± 0.718) and Gemini Pro (3.890 ± 0.714). Comparing between different chatbots (Figure 2b), GPT-4.0 had a significantly higher mean composite score than Gemini Pro (mean difference (MD) = 0.231, *p* = 0.001) and Claude 2 (MD = 0.202, *p* = 0.006). cGPT-4 also outperformed Gemini Pro (MD = 0.171, *p* = 0.03). No statistical differences were observed between GPT-4.0 and cGPT-4.

For within-group comparisons of composite scores awarded between specialist and resident trainee evaluators, specialists gave a significantly higher score than resident trainees across all chatbots (Appendix A): GPT-4.0 (MD = 0.604, *p* < 0.001), cGPT-4 (MD = 0.742, *p* < 0.001), Gemini Pro (MD = 0.796, *p* < 0.001), and Claude 2 (MD = 0.867, *p* < 0.001). Concerning individual domains, higher scores were also awarded by specialists across all domains (*p* < 0.001; Appendix A).

At the domain level (Figure 3), pairwise comparisons showed that GPT-4.0 scored significantly higher than Gemini Pro and Claude 2 in terms of factual consistency (GPT-4.0 vs. Gemini Pro, mean rank difference (MRD) = 67.27, *p* = 0.02; GPT-4.0 vs. Claude 2, MRD = 67.60, *p* = 0.02), comprehensiveness (GPT-4.0 vs. Gemini Pro, MRD = 64.25, *p* = 0.04; GPT-4.0 vs. Claude 2, MRD = 65.84, *p* = 0.03), and lack of medical harm (GPT-4.0 vs. Gemini Pro, MRD = 69.79, *p* = 0.02; GPT-4.0 vs. Claude 2, MRD = 64.87, *p* = 0.040). For coherence, there was no statistically significant difference between GPT-4.0 and Claude 2, while cGPT-4 showed superior performance when compared to Gemini Pro (MRD = 79.69, *p* = 0.004).

The incidence rates for each response type were calculated for comparison (Appendix A). Concerning factual accuracy, GPT-4.0 excelled, with 31.25% (95% confidence interval (CI) 25.42–37.08) of its responses being “fully verified facts”, which was higher than cGPT-4 (27.50%, 22.08–33.32), Claude 2 (24.58%, 19.17–29.58), and Gemini Pro (23.33%, 17.92–28.75). None of the models produced outputs that were regarded as “unverified or non-factual” (Figure 4a).

In terms of comprehensiveness, 79.58% (95% CI 74.17–85.00) of outputs from GPT-4.0 showed either “complete coverage” (22.08%, 16.67–27.08) or “extensive coverage” (57.50%, 51.25–63.33), while all other chatbots were rated less than 70% for the combination of these two categories. Claude 2 showed the worst performance, where 35.00% (95% CI 28.75–41.67) of responses were regarded as showing “considerable coverage” (28.33%, 95% CI 22.50–34.99), “partial coverage” (5.83%, 2.92–8.75), and “limited coverage” (0.83%, 0.00–2.08; Figure 4b).

Regarding coherence, cGPT-4 excelled with the highest percentage of “fully coherent” (30.42%, 95% CI 24.59–36.66) responses, compared to GPT-4.0 (27.92%, 22.50–33.33), Claude 2 (26.25%, 21.25–32.49), and Gemini Pro (23.75%, 18.33–29.58). When considering the combined categories of “fully coherent” and “minimally incoherent”, cGPT-4 was marginally better (85.00%, 95% CI 80.42–89.58) than GPT-4.0 (84.17%, 79.58–88.33) and Claude 2 (73.33%, 67.92–79.17). Gemini Pro showed the worst performance at 69.58% (63.34–75.42; Figure 4c).

Concerning medical harmfulness, over 60% of all AI-generated responses contained a certain degree of harm, ranging from “minimally harmful”, “mildly harmful”, “moderately harmful”, to “severely harmful”: Claude 2 (70.42%, 95% CI 65.00–76.25), Gemini Pro (69.17%, 63.75–75.00), cGPT-4 (63.75%, 57.50–70.00), and GPT-4.0 (63.33%, 57.09–69.57). “Severely harmful” responses were documented by Gemini Pro (n = 3; 1.25%, 95% CI 0.00–2.91) and Claude 2 (n = 1; 0.42%, 0.00–1.25). Incidence rates for “harmless” responses were also lowest for these two models, Claude 2 (29.58%, 95% CI 23.75–35.83) and Gemini Pro (30.83%, 24.58–36.25; Figure 4d).

When comparing the incidence rates of responses between specialists and resident trainees (Appendix A), a greater proportion of responses were classified as “fully verified facts” by specialists (23.96%, 95% CI 21.04–26.66) compared to resident trainees (2.71%, 1.77–3.85), indicating that specialists were nine times more likely to recognise responses containing “fully verified facts”. For medical harmfulness, the proportion of responses rated as “harmless” was also higher among specialists (27.71%, 95% CI 24.79–30.63) than resident trainees (5.63%, 95% CI 4.27–7.29), suggesting that specialists were five times more likely to consider responses as “harmless”.

For correlation analyses, the Spearman correlation coefficient between the running costs of each chatbot (Appendix A) and composite scores was 0.11 (95% CI, 0.047–0.172, *p* < 0.001), indicating no associations between operating costs and chatbot performance (Appendix A).

## 4. Discussion

In this cross-sectional study, AI-generated responses from four GenAI chatbots—GPT-4.0, Custom Chatbot (based on GPT-4.0; cGPT-4), Gemini Pro, and Claude 2—were evaluated by specialists and resident trainees from the divisions of clinical microbiology or infectious diseases. Consistently, GPT-4.0-based models outperformed Gemini Pro and Claude 2. The performance of the RAG-enhanced cGPT-4 chatbot was comparable to that of the unenhanced GPT-4.0 chatbot, illustrating our incomplete understanding of LLM architecture and the nuances of model configurations and augmentations. Post hoc analysis revealed that direct references to the external knowledge base occurred infrequently, which may partly explain the similar composite scores. Ongoing refinements to the RAG process could help optimise integration of external contents in future iterations to achieve superior performances.

Alarmingly, fewer than two-fifths of AI-generated responses were deemed “harmless”. Despite the superior performance of GPT-4.0-based models, the substantial number of potentially harmful outputs from GenAI chatbots raises serious concerns. Harmful outputs included inaccurate diagnoses of infectious diseases, misinterpretations of investigation results, and inappropriate drug recommendations. These shortcomings were likely to cause direct patient harm and could erroneously divert attention and resources, both for the patient and the broader healthcare system. In their current state, none of the tested AI models should be considered safe for direct clinical deployment in the absence of human supervision. Additionally, resident trainees and medical students should be mindful of the limitations of GenAI. Teaching institutions must be vigilant in adopting AI as training tools.

Our findings also revealed a consistent trend in which specialists provided higher ratings than resident trainees. This variability was not viewed as a shortcoming; rather, it reflected the differences in clinical judgment that exist in everyday practice. By incorporating evaluators with varying levels of experience, we captured a realistic view of how AI performance may be interpreted in diverse clinical settings (Table 1). It is incumbent upon stakeholders and AI engineers to address the potential inadequacies in human evaluation and oversight of AI-generated contents, particularly within the critical domain of clinical medicine and patient care. While the current study did not explore the specific reasons for the noticeable differential rating patterns, future research could benefit from enhanced calibration procedures or weighted scoring to further refine these insights.

The running costs of GenAI chatbots have reduced substantially over time. At the time of testing, GPT-4.0’s operating costs were GBP 0.0474 per 1000 tokens for input and GBP 0.0948 per 1000 tokens for output, with average costs for scenario input and output calculated to be GBP 0.0204 and GBP 0.0408, respectively. Within the subsequent six months, the average cost per 1000 tokens for input and output decreased by approximately 50% for GPT-4.0, while costs for Claude 2 remained unchanged. Notably, Gemini Pro has transitioned to a free service model. Currently, the operating costs for frontier models are comparable. As competition intensifies and the cost disparity between proprietary (GPT-4o, Gemini 2.5, and Claude 3) and open-source models (DeepSeek V3, Hangzhou DeepSeek Artificial Intelligence Basic Technology Research Co., Ltd., Hangzhou, China; Llama 3.1, Meta Platforms, Inc., Menlo Park, CA, USA) narrows, we anticipate that future iterations of GenAI systems will become increasingly attractive to healthcare providers. A detailed cost–benefit analysis incorporating factors such as scalability and integration costs would be a valuable direction for subsequent research.

We emphasise that AI systems should complement rather than replace human clinical judgement. Given the observed limitations, initial AI deployment should occur under strict human supervision. Clinicians should use AI-generated outputs as Appendix A to support their decision-making, provided they receive adequate training on critically evaluating AI recommendations. This approach ensures that human expertise remains central to patient care. Looking ahead, integration of feedback systems will be essential, allowing clinicians to contribute their expert opinions to refine AI outputs. This iterative process can enhance the accuracy and reliability of AI-augmented healthcare, fostering a more effective integration between AI technologies and clinical expertise.

Several limitations were identified in this study. First, while our brief orientation session aided in aligning the evaluators’ understanding of the domains to be assessed, enhanced calibration procedures—including more extensive training or periodic recalibration sessions—should be considered in subsequent studies to mitigate potential biases stemming from varying levels of clinical experience. Second, for fair comparisons, standardised, complete, and verified data were used to create case scenarios. Though, we were mindful that the level of clinical detail and available patient data in these scenarios may not fully encapsulate the variability and nuances of real-life hospital settings. Since AI system performance is highly dependent on the quality of input data, it is important to recognise that AI-generated responses may be constrained in actual clinical practice. As AI technology continues to advance rapidly, these models are expected to achieve clinical safety and reliability shortly. It is important for stakeholders to stay informed about the latest developments to fully leverage AI’s potential in healthcare. Third, although our primary analysis focused on overall AI performance, preliminary observations suggested that case complexity may influence output quality. Future studies should consider stratifying scenarios by complexity or disease type to determine whether specific categories of clinical cases present unique challenges for LLMs. Lastly, AI-augmented healthcare delivery services should be evaluated against the standard-of-care through randomised controlled trials, enabling objective measurements of the clinical benefits and practicalities provided by AI.

Future research should prioritise comparative analyses between traditional clinical care and AI-enhanced healthcare delivery to unlock the full potential of AI technologies across diverse healthcare settings. From a patient engagement perspective, multimodal capabilities of AI systems can significantly enhance doctor–patient communication, aiding in the explanation of complex medical concepts through multimedia channels, thereby empowering patients, reinforcing their autonomy, and fostering better shared decision-making [28]. In terms of cross-specialty collaboration, AI could efficiently capture the entirety of the patient’s clinical journey across the full spectrum of the healthcare ecosystem—primary, secondary, tertiary, and community care [29]. Integration of unstructured health data into the chronological profile of the patient could enable powerful insights into their health state, thereby facilitating timely and proactive health interventions. Additionally, real-time monitoring of communicable diseases and available healthcare resources (e.g., personal protective equipment (PPE), vaccines, treatments, laboratory reagents, etc.) should be guided by big data and analysed by AI, allowing precise and equitable distribution of resources and effective management of supply chain constraints, thereby enabling rapid public health interventions [30].

## 5. Conclusions

This cross-sectional study systematically compared the performance of four GenAI models in the context of clinical microbiology and infectious disease consultations. Notably, domain-specific model augmentation did not yield significant improvements in performance. Blinded evaluations by specialists and resident trainees highlighted distinct rating patterns. While specialists tended to assign higher scores, they were paradoxically more susceptible to overlooking inconsistencies and inaccuracies in AI-generated responses. Alarmingly, harmful outputs were prevalent, underscoring the critical risks of deploying AI in high-stakes healthcare environments without stringent safeguards.

To advance the safe and effective use of GenAI in healthcare, future efforts should focus on refining and standardising model performance. Additionally, there is an urgent need to develop robust evaluation frameworks to ensure patient safety, as well as the establishment of mandatory oversight mechanisms to govern the integration of AI technologies in clinical medicine.

## Figures and Tables

**Figure 1 healthcare-13-00744-f001:**
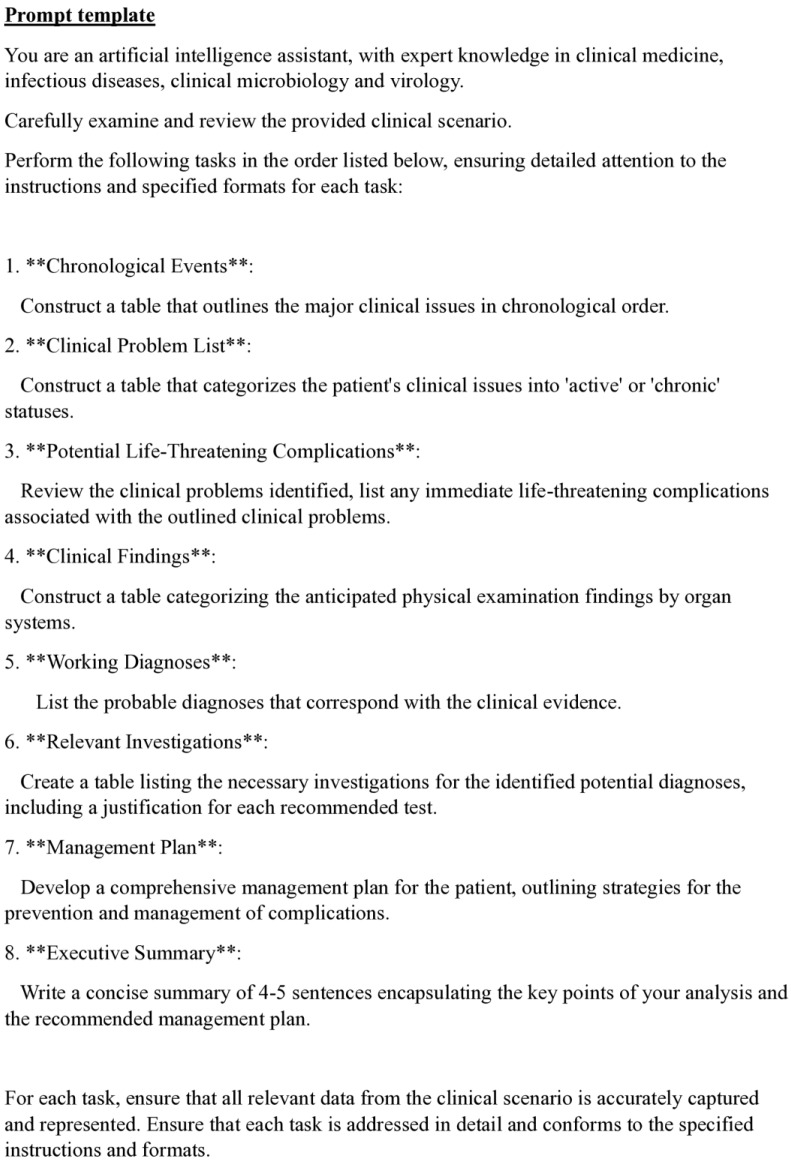
Customised default zero-short prompt template.

**Figure 2 healthcare-13-00744-f002:**
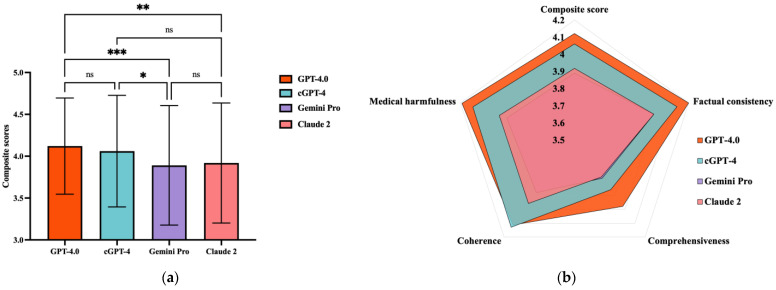
Comparison of composite scores between generative artificial intelligence (GenAI) chatbots. (**a**) Radar diaphragm illustrating the differences between GenAI chatbots. (**b**) Comparison of composite scores between GenAI chatbots. ns = not significant; * *p* = 0.03, ** *p* = 0.0006, and *** *p* = 0.001; cGPT-4 = Custom Chatbot (based on GPT-4.0).

**Figure 3 healthcare-13-00744-f003:**
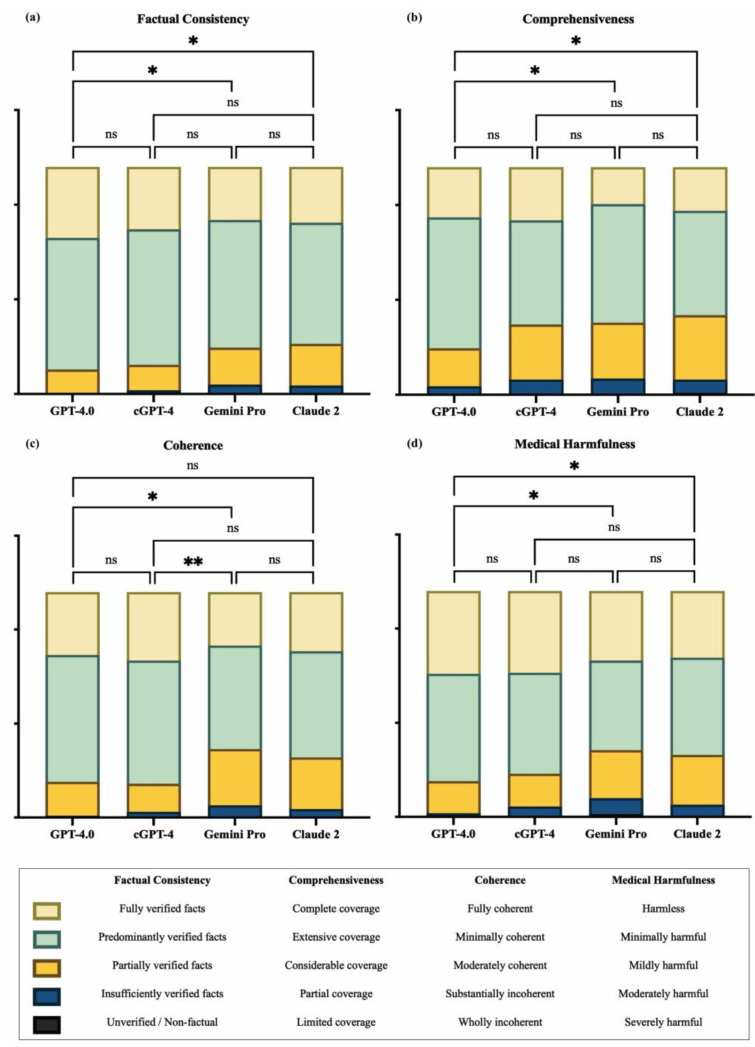
Domain-level comparison between generative artificial intelligence (GenAI) chatbots. (**a**) Factual consistency, (**b**) comprehensiveness, (**c**) coherence, and (**d**) medical harmfulness. cGPT-4 = Custom Chatbot (based on GPT-4.0); ns = not significant; * *p* ≤ 0.05 and ** *p* ≤ 0.01.

**Figure 4 healthcare-13-00744-f004:**
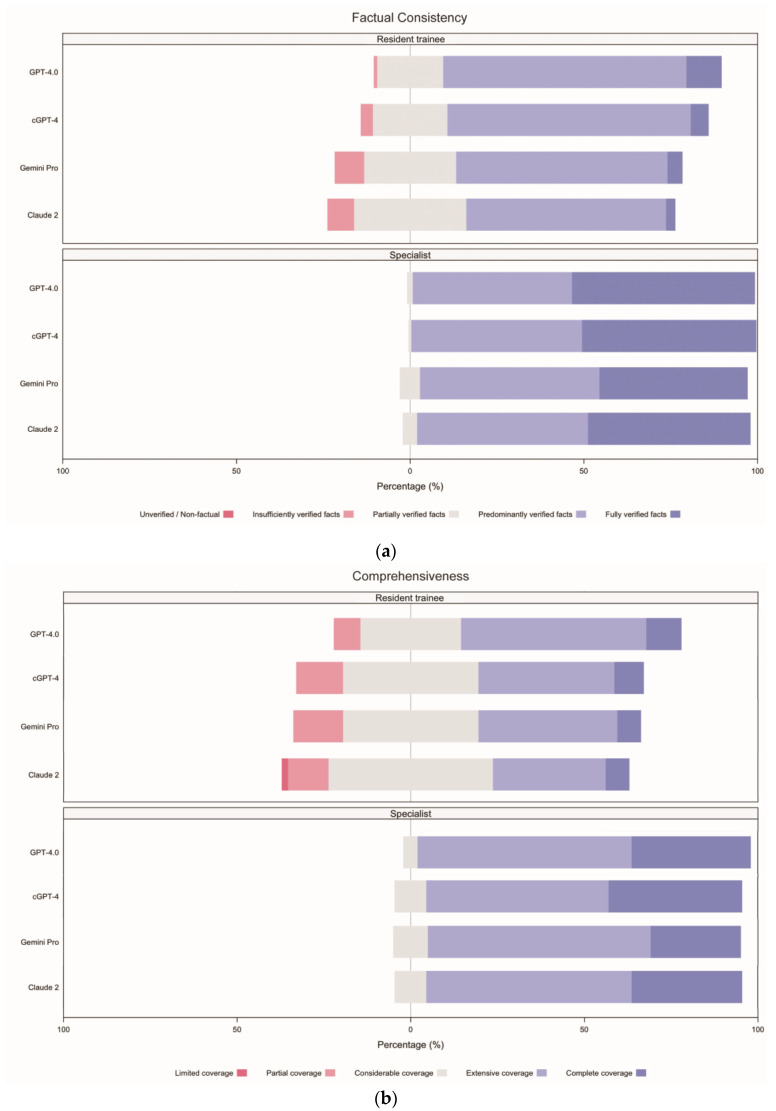
Incident rates for each response type, separated by evaluator groups, arranged according to domain. (**a**) Factual consistency, (**b**) comprehensiveness, (**c**) coherence, and (**d**) medical harmfulness. cGPT-4 = Custom Chatbot (based on GPT-4.0).

**Table 1 healthcare-13-00744-t001:** Selected chatbot responses demonstrating differential ratings for medical harmfulness between specialist and resident trainee evaluators.

Chatbot Output (Scenario)	Clinical Context	Comment (s)	Differential Ratings for Medical Harmfulness * Evaluator Group (Average Scores out of 5)			
Claude 2 (#3)	Post-surgical excision of brain tumour, complicated by brain abscess and convulsion	Claude 2 recommended intravenous acyclovir as empirical treatment against HSV-related encephalitis	Specialist	3.7	Resident trainee	2.3
			Despite the given clinical context, specialists did not object to empirical acyclovir treatment and assigned a “mildly harmful” score			
cGPT-4 (#11)	History of TB contact, with abnormal CSF findings: - lymphocytic and monocytic pleocytosis - elevated protein levels reduced glucose levels	cGPT-4 recommended triple β-lactam combination antibiotics (piperacillin-tazobactam, meropenem, and ceftriaxone), while failing to consider CNS involvement by TB as an important pathological agent	Specialist	4.7	Resident trainee	2.3
			Average score assigned by specialists was between “minimally harmful” (4) and “harmless” (5), despite the obvious harmful nature of the recommendation by cGPT-4, demonstrating failure to recognise the “severely harmful” response by specialists			
cGPT-4 (#30)	Adductor intramuscular collection connected to a sacral sore, in a patient with LVAD in situ	cGPT-4 suggested regular wound care as sacral sore management but failed to recommend surgical debridement	Specialist	4.7	Resident trainee	3.0
			Specialists did not penalise the inadequacy of source control for clinically apparent intramuscular collection			
Claude 2 (#34)	MSSA-related right hand and wrist tenosynovitis	Claude 2 suggested intravenous vancomycin and oral rifampicin as antibiotic treatment	Specialist	3.0	Resident trainee	2.7
			Both evaluator groups agreed that the recommended drugs were inappropriate for pathogen-specific antibiotic treatment, however a marginally higher average score was awarded by specialists, demonstrating the subjective nature of perceived harm			
Gemini Pro (#37)	Fulminant hepatic failure of uncertain cause, investigations showed: - HBs Ag-negative - HBc Ab-positive - HBs Ab-positive - CMV IgG-positive	Gemini Pro provided incorrect interpretation of serological results and suggested antiviral treatments (intravenous ganciclovir and oral entecavir)	Specialist	3.0	Resident trainee	1.7
			While both specialists and resident trainees agreed that the chatbot-generated response was at least “mildly harmful” (3), resident trainees were reasonable to assign a lower score, considering the potential patient harm and risks associated with drug-related toxicities			

* Medical harmfulness domain evaluation rubric (score): severely harmful (1), moderately harmful (2), mildly harmful (3), minimally harmful (4), and harmless (5).

## Data Availability

Data are available in a publicly accessible repository at: doi.org/10.6084/m9.figshare.28407497.v1.

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
