# Peer review of "Generative Artificial Intelligence Models in Clinical Infectious Disease Consultations: A Cross-Sectional Analysis Among Specialists and Resident Trainees"

_healthcare, 2025, doi:10.3390/healthcare13070744_

Round 1
Reviewer 1 Report
Comments and Suggestions for Authors
I have only one question, but that may lead to massive work or may need a small clarification.
The temperature will control the randomness of the output, which a higher temperature may lead to more diverse and creative response. I can see that the Claude 2 was set to 0.5, the GPT was set to 0.35, and the Gemini Pro is unknown. Although the values are recommended by Poe, for a certain experiment, should you set them to the same level to compare the results? That may also lead to an interesting question: How much will the temperature lead to the difference? Furthermore, are the differences between Claude 2 and GPT caused by the different default settings?
One small comment, line 364 may change to "This study" or "We".
Reviewer 2 Report
Comments and Suggestions for Authors
Methods and Results section
Aside from internal consistencies across chatbots, the author should add information about questionnaire internal consistency (item question in domain factual consistency, comprehensiveness, and the absence of medical harm) and baseline characteristic of respondent (specialist and resident evaluator). What is "Insufficiently verified facts" in the factual consistency domain, or “incoherence” in the coherence domain? For example: meaning of each Likert scale for ‘coherence’ (how logically structured and clinically impactful the chatbot responses were).
I think that specialist or resident training background will affect their rewarding scale. How long and what is their previous experiment with chatbot?
Comments on the Quality of English LanguageModerate
Reviewer 3 Report
Comments and Suggestions for Authors
Dear Authors
First of all, I would like to state that your study is quite successful in terms of its in-depth analysis and statistical robustness regarding the use of GenAI models in clinical infectious disease consultancy. Your research design is methodologically strong, the evaluation process is well-structured, and the statistical methods used increase the reliability of the study. In particular, revealing the evaluation differences between specialists and assistant doctors provides an important perspective on how AI affects the human factor in medical practice.
Your study is largely suitable for publication, and can be made stronger by making improvements only at certain points. First of all, the introduction section can be expanded. Although the literature review is generally sufficient, there is a lack of comparison with medical AI models such as Med-PaLM 2 and MEDITRON. It would be useful to add a brief explanation of why these models were not included in the study or the main differences between their working principles and the tested models. In addition, if more context is provided regarding the regulations of AI in the health field (for example, the views of institutions such as FDA, EMA, WHO), the place of the article in medical decision support systems can be better positioned.
The details of the research environment would also be beneficial for the generalizability of the study. A better explanation of how the clinical scenarios were selected, the size of the hospital environment, and the variety of patients could help readers better understand the scope of the study. Similarly, a little more detail on the operating parameters of the AI ​​models used (e.g. temperature and other model settings) would be especially useful for researchers interested in optimizing AI systems.
The results are generally presented in a clear and powerful way, with graphs and tables organized in an understandable manner. The supplementary material is particularly commendable in detail. Sharing the datasets and additional analyses in detail increases the transparency of the study and contributes to the wider academic community. These supplementary materials stand out as an important component that increases the reliability and reproducibility of your study.
The main findings of your study indicate that GenAI is currently not safe to integrate directly into clinical applications and requires expert supervision. These results are in line with the existing literature on the use of medical AI and contribute to an important discussion on how to make AI more reliable in clinical applications. Adding more comments on how your findings can be integrated into clinical practice can make the study more instructive.
Overall, the structure and content of the article do not require major changes. Only some minor improvements on the points mentioned above can make it more effective and appealing to a broad academic audience. As such, your study provides an important academic contribution to the applicability of AI in healthcare and has important implications for the future of medical decision support systems.
Reviewer 4 Report
Comments and Suggestions for Authors
This study evaluates the potential of generative artificial intelligence (GenAI) models in clinical consultation services within the field of infectious diseases and clinical microbiology. GPT-4.0, a specialized GPT-4.0-based chatbot (cGPT-4), Gemini Pro, and Claude 2 models were compared across 40 different clinical scenarios, with their responses assessed by experts and resident doctors in terms of accuracy, comprehensiveness, consistency, and medical safety. While the study provides significant insights into the clinical integration of GenAI, it also highlights that current models are not yet safe for direct clinical applications. From a scientific perspective, the study is unique in offering a methodological framework for evaluating LLM-based systems in healthcare and analyzing the impact of expert evaluation processes.
**Limitations and Recommendations:**
- The study only assessed 40 clinical scenarios, which may be insufficient to draw generalizable conclusions about model performance. Researchers could consider expanding the sample size in future studies.
- The AI-generated responses were not directly compared to real clinical decisions. A direct comparison with human doctors' decisions would help determine the practical clinical usability of AI.
- Factors influencing chatbot performance (e.g., prompt engineering, variations in training data) should be explored in greater detail.
- More than 60% of AI-generated responses were reported to contain some degree of medical harm, but the potential clinical consequences of these inaccuracies were not thoroughly discussed.
- While the study emphasizes that current AI models should not be used directly in clinical settings, providing practical guidance on how they could be safely integrated would add value to the article.
- Discussing how to make AI outputs safer and establishing a structured process for human oversight would be beneficial.
Round 2
Reviewer 1 Report
Comments and Suggestions for Authors
No comments at all.